# Burnout in Nursing Managers: A Systematic Review and Meta-Analysis of Related Factors, Levels and Prevalence

**DOI:** 10.3390/ijerph17113983

**Published:** 2020-06-04

**Authors:** María José Membrive-Jiménez, Laura Pradas-Hernández, Nora Suleiman-Martos, Keyla Vargas-Román, Guillermo A. Cañadas-De la Fuente, José Luis Gomez-Urquiza, Emilia I. De la Fuente-Solana

**Affiliations:** 1Ceuta University Hospital, National Institute of Health Management, Loma del Colmenar s/n, 51003 Ceuta, Spain; mariajose.membrive@gmail.com; 2Virgen de las Nieves University Hospital, Andalusian Health Service, Avenida del Sur N. 11, 18014 Granada, Spain; lauraphl9@gmail.com; 3Faculty of Health Sciences, University of Granada, Cortadura del Valle s/n, 51001 Ceuta, Spain; norasm@ugr.es; 4Faculty of Psychology, University of Granada, Campus Universitario de Cartuja s/n, 18071 Granada, Spain; keyvarom@ugr.es; 5Faculty of Health Sciences, University of Granada, Avenida Ilustración, 60, 18016 Granada, Spain; gacf@ugr.es; 6Brain, Mind and Behaviour Research Center (CIMCYC), University of Granada, Campus Universitario de Cartuja s/n, 18071 Granada, Spain; edfuente@ugr.es

**Keywords:** burnout, nursing management, occupational health, prevalence, risk factors

## Abstract

Burnout syndrome is a major problem in occupational health, which also affects nursing managers. The main aim was to analyze the level, prevalence and risk factors of burnout among nursing managers. A systematic review with meta-analysis was conducted. The databases used were Medline (Pubmed), PsycINFO, CINAHL, LILACS, Scielo and Scopus. The search equation was “burnout AND nurs* AND (health manager OR case managers)”. Nursing managers present high levels of emotional exhaustion and a high degree of depersonalization. Some studies show that variables like age, gender, marital status, having children or mobbing and other occupational factors are related with burnout. The prevalence estimation of emotional exhaustion with the meta-analysis was high; 29% (95% CI = 9–56) with a sample of n = 780 nursing managers. The meta-analytical estimation of the correlation between burnout and age was *r* = −0.07 (95% CI = −0.23–0.08). Work overload, the need to mediate personnel conflicts, lack of time and support from superior staff, contribute to the development of burnout among nursing managers.

## 1. Background

In recent years, levels of burnout syndrome have raised, a phenomenon that has drawn the attention of both academics and industry professionals [1]. Accordingly, the problem must be analyzed from diverse perspectives [2]. The term ‘burnout’ was coined in the 1970s to describe the physical and emotional exhaustion that workers may experience on the job, especially those who provide some type of service to others [3]. 

Burnout appears when workers are continuously exposed to stressors that they cannot face. This makes them feel exhausted, lacking in energy and mentally fatigued [1]. This syndrome is highly prevalent among health professionals, especially among nursing staff [4]. Numerous studies have evidenced nurses´ vulnerability to the effects of burnout [5,6]. 

Among nurses, and other workers, burnout is characterized by the appearance of the following effects: emotional exhaustion (EE), resulting in a progressive loss of energy; depersonalization (D), in the form of negative attitudes towards patients and coworkers; feelings of low personal accomplishment (PA) or loss of confidence [7]. Nursing is considered a particularly stressful profession due to the proximity to patients and the strong emotional involvement that this often produces [8]. Nurses with burnout usually have a negative view of the health institution and, among other consequences, there is greater absenteeism, a rise in staff turnover, a worsening of the work environment and strained relationships among workers [9]. This situation impacts on the quality of care and consequently is detrimental for all users of healthcare services [10].

On the other hand, nursing managers are leaders who guide nurses in their department and help them to adopt new ideas and practices for the improvement of the units. Moreover, nurse managers act as the middleman who communicate with both staff and upper-level management. They are primarily responsible for overseeing the staff, managing the finances and records within the nursing department and to providing the best care possible to patients if necessary.

Every day, nursing managers are exposed to demanding situations and stress, often requiring extended workdays [11]. In addition, the profession assigns high levels of social responsibility. Every single unit’s trouble must be addressed by these managers, who very often experience overload [4], and find it difficult to reconcile work and family life [12].

The main signs and symptoms of burnout among nursing managers include tiredness, difficulty in concentrating, poor organization, a greater number of errors, decreased quality of work, a lack of energy (both at work and elsewhere), anxiety and frustration [8]. Many factors are associated with the appearance of burnout syndrome, but time pressures and the existence of conflicts in the workplace are especially important as sources of increased stress and related effects. When these circumstances persist over a significant period, they can lead to the appearance of burnout [13].

Nursing managers are also important figures in helping avoid the burnout of the nursing staff from different units [4], and their daily practice is different than the daily practice of the clinical nurses. So, analyzing if nursing managers are affected by burnout, what the possible risk factors are, and estimating the prevalence for them as has been done with nurses from different units [4,6], is important. 

Thus, the question and aim for this review was what are the levels, risk factors and the prevalence of burnout in nursing managers? 

## 2. Methods

### 2.1. Data Sources, String and Inclusion Criteria

A systematic review and meta-analysis were performed, in accordance with the PRISMA guidelines. 

The first step in this process was to search the following electronic databases: Medline, PsycINFO, CINAHL, LILACS, Scielo and Scopus.

The string used was “burnout AND nurs* AND (health manager OR case managers)”. No restrictions were used on the publication date, study methods or sample size.

The second step was to examine and assess all the studies and reviews obtained on this topic, and finally, to perform the same operation for all the references cited in these studies.

The following inclusion criteria were applied: (a) studies published a maximum of five years ago; (b) studies that analyzed a sample of nursing managers; (c) studies that measured burnout in this sample and specified in their results the number or proportion of persons with burnout; (d) the language used in the document (Spanish, English, Portuguese or Italian); (e) use of the Maslach Burnout Inventory (MBI) or similar (MBI is a psychological inventory consisting of 22 items, which measure the different dimensions of Burnout, EE and PA. It takes between 10 and 15 min to fill out). 

The exclusion criteria were: (a) samples of nurses not classed as managers; (b) studies with mixed samples lacking independent information for nursing managers; (c) qualitative studies; (d) studies with low methodological quality (studies with more than 3 negative responses to Ciapponi´s critical read checklist) [14].

### 2.2. Coding of Results

Two members of the team, independently, performed the search, selection and detailed reading of the publications. In cases of disagreement, other researchers of the group were consulted as well.

The following variables were considered and recorded in a specific data definition record table (see Table 1): (a) authors; (b) year of publication; (c) country of publication; (d) language; (e) sample size; (f) percentage of nursing managers in the sample; (g) sex and age of the sample population (mean, standard deviation, median or range).

Methodological variables: (a) burnout measurement instrument; (b) in the case of the Maslach Burnout Inventory (MBI), subtype specification (MBI, MBI-Human Services, MBI-General Services); (c) use of the original instrument or an adapted version of it.

Variables reflecting the impact of burnout: Main outcomes of the presence of burnout in nursing managers. Levels of burnout (low, medium or high) for each dimension as established in the MBI.

### 2.3. Data Analysis

A descriptive analysis was made of the study variables thus included, concerning the quality of each publication selected, according to the levels of evidence and the degrees of recommendation proposed by the Working Group regarding evidence levels of the Oxford Centre for Evidence-Based Medicine (OCEBM).

The StatsDirect software, a statistical software package for general health science users, was used to analyze the main data. Firstly, a sensitivity analysis was carried out and the publication bias was assessed using the Egger’s linear regression test. Two random effects meta-analysis were done to calculate the estimations and their confidence intervals: one for the correlation between age and burnout and one for the burnout prevalence. The heterogeneity analysis of the sample was carried out with the *I*^2^ index. The cut-off points of the MBI were used in the included studies for knowing the prevalence of high emotional exhaustion. That information was used for the meta-analysis prevalence estimation.

## 3. Results

### 3.1. Search Description and Studies Included

The search was conducted in March 2019. The databases showed 438 studies, which were reduced to 25 after reading the title and abstract. Then after applying the corresponding inclusion and exclusion criteria, n = 11 studies were finally selected, excluding 10 articles for not being related with the study topic and 4 for not having data (Figure 1). As well as these, 14 studies were excluded due to their low methodological quality. 

Over half of these studies were published between 2014 and 2019, describing research conducted in Canada, Japan, Switzerland, Belgium, USA and Finland.

The sample population was composed of nursing managers working in hospitals, most of which were female (women accounted for over 90% of the study population in most cases) [15,16]. The average age of the nurses was about 45 years. For most of them, the researchers used the MBI test to measure the different dimensions of the burnout syndrome in the sample, except for one study selected, in which the researchers used the Japanese Burnout Inventory [15] (Table 1).

### 3.2. Levels and Prevalence of Burnout among Nursing Managers

Burnout is a frequent risk among workers in many areas, including health care [23]. Studies have shown that nursing managers are especially exposed to emotional exhaustion (EE) [15,16,18,19], which is closely related to high levels of occupational stress [21] and low levels of assertiveness [17].

Depersonalization (D) is considered less virulent than EE, but when it appears, it is often apparent as insensitivity towards co-workers and patients, hampering relationships between managers and other employees [18].

Low levels of personal accomplishment (PA) reduced effectiveness, heightened the difficulties experienced and reduced individuals’ interest in their work [18]. This dimension of burnout was especially likely to occur when workers were subjected to overload and had a high degree of responsibility. This was often the case for nursing managers, who thus presented a strong predisposition towards problems in this respect [15].

### 3.3. Risk Factors and Mediating Factors for Burnout in Nursing Managers

Among other sociodemographic factors, studies have shown that burnout was related to age [9,15,17,18,19,20] and sex [9,16,18,19,20]. Thus, the risk of developing burnout was especially acute among women aged 40−50 years old. Other mediating factors were marital status and maternity; most of the affected sample were married [17,18,21,23] and had children [17,18]. Furthermore, full-time workers were more likely to suffer burnout than those working part time [9,15]. It has been observed that failing to receive the necessary support from co-workers, subordinates and superiors had a strong negative impact on nursing managers and was directly associated with the rejection of their role as a nursing manager [9]. Some studies had even observed the existence of mobbing (behaviour of people harming others with whom they work), which also aggravates burnout and causes premature abandonment [23].

### 3.4. Meta-Analysis Results

No publication bias was found, the Egger’s analysis being non-significant. Regarding the sensitivity analysis, the meta-analytical estimation values did not vary when eliminating any of the studies.

In the analysis of heterogeneity, the *I*^2^ index was 48.1% for the correlation meta-analysis and 96.6% for the prevalence meta-analysis. The meta-analytical estimate of the correlation between burnout and age was *r* = −0.07 (95% CI = −0.23–0.08), indicating that when age is lower burnout levels are higher (Figure 2).

With regards to emotional exhaustion prevalence, the meta analytical estimation was 29% (95% CI = 9–55) with a sample of n = 780 nursing managers, as shown in Figure 3. There were not enough studies for depersonalization (D) and personal accomplishment (PA) meta-analysis.

## 4. Discussion

Burnout syndrome is generally considered to be composed of three dimensions: emotional exhaustion (EE), depersonalization (D) and personal accomplishment (PA). It has been argued that nursing managers are particularly vulnerable to EE, largely due to the work overload they often experience. This may be caused by staff shortages, which makes it difficult to provide an adequate response to the demands for care [26,27]. This problem, together with the quite common lack of assertiveness with managers, regarding the perceived lack of support, is associated with great dissatisfaction among these professionals [26,28] and high levels of stress [27,29]. In turn, this aggravates EE and hence burnout.

Depersonalization (D) is also ordinary among these professionals, leading them to avoid contact with other employees. This fact can end in serious problems of insensitivity towards patients and co-workers, affecting relationships and worsening both the work environment and the quality of work performed. This manifestation of burnout is a consequence of the individual attempting to adapt to the situation and to alleviate the tension experienced in the workplace. However, it is perceived by others as a lack of leadership [26,30].

Some studies have ascribed low personal accomplishment (PA) to feelings of unease and disconnection, from the workplace and from co-workers. Feelings of this type may be due to a lack of involvement with work and its responsibilities, which can impair performance and productivity [31]. The high level of responsibility imposed aggravates this situation, with managers feeling themselves under great pressure, with too much to do and little time to do it [32]. 

Among other risk factors for burnout, being exposed to situations in which the demands placed on the individual exceed their ability to cope with them will obviously endanger the sense of well-being [33]. Such situations arise when an excessive number of patients must be cared for, when there are too few personnel or when nursing managers are called upon to exercise skills beyond their command [34,35]. Furthermore, overload leads to a loss of productivity, where problems arise in performing the activities required, and problems such as concentration deficit or difficulty in meeting deadlines will increase the number of errors made [36], producing a direct and immediate prejudice towards the nurses for whom the manager is responsible, and therefore towards their patients. In addition to the above problems, nursing managers must address and resolve the conflicts that may arise among their staff. It has been observed that promoting a healthy working environment and resolving conflicts requires constant effort [30,37]. This obligation, together with the perceived lack of support from superiors, co-workers and subordinates [22], makes these managers very susceptible to burnout. Furthermore, there are not enough studies that measure burnout in physician managers, therefore no comparisons can be made with our study population [38].

Nursing managers often bear a heavy workload, and this is a major risk factor for burnout. In addition to providing nursing care, they must deal with the personnel in their charge, resolve conflicts [18], establish productive relationships with other personnel—both within the hospital and elsewhere—and make good use of health and social resources [15,24].

When nursing managers experience burnout, they cannot properly meet the needs of their patients, which results in lost productivity and feelings of dissatisfaction [15,23].

Moreover, burnout among nursing managers seems to be related to its appearance among other members of staff. Therefore, in order to maintain a healthy working environment and to preserve the mental welfare of hospital staff, all possible effort should be made to minimize levels of burnout among nursing managers [20]. To achieve this, possible interventions that may be effective could include senior officials lending their support and help to lighten managers’ responsibilities, especially those of an administrative nature, for example, by providing more auxiliary staff [9]. In addition, studies have shown that mindfulness-based interventions can significantly alleviate the degree of burnout experienced by nursing managers [24]. Access to adequate resources, both material and immaterial (such as time), may also help burnout prevention [23].

Furthermore, various methods have been proposed to address or prevent this problem, since nursing managers either directly or indirectly exert a major influence on the quality of care provided to patients [29,39]. It has been suggested that changes should be made in how duties are organized. An example of this would be establishing rotations, so that workers do not spend too much time in the same work space, and providing more support in the work environment [40,41], which is believed to reduce burnout by strengthening relationships within the healthcare team and by generating positive attitudes towards patients, thus improving the well-being of participants in these activities [42,43].

This study has some limitations. First, the number of included studies is not high and the studies have been done in different countries with different healthcare systems. Thus, some results should be considered with caution, and more research should be done focusing on this topic in the future.

## 5. Conclusions

The results of the study help to describe how burnout syndrome affects nursing managers and their risk factors. This meta-analysis paper is useful for recognizing mental health challenges for nursing managers and may increase awareness of the prevalence and impact of burnout in nursing managers. This should promote the prevention and treatment of burnout in these healthcare workers. Moreover, the manuscript, based on scientific evidence, promotes the use of new strategies to prevent burnout in nursing managers in the hospital environment. Among nursing managers, those most vulnerable to burnout may have the following characteristics: women, aged 40–50 years, working full time, being married and having children. 

## Figures and Tables

**Figure 1 ijerph-17-03983-f001:**
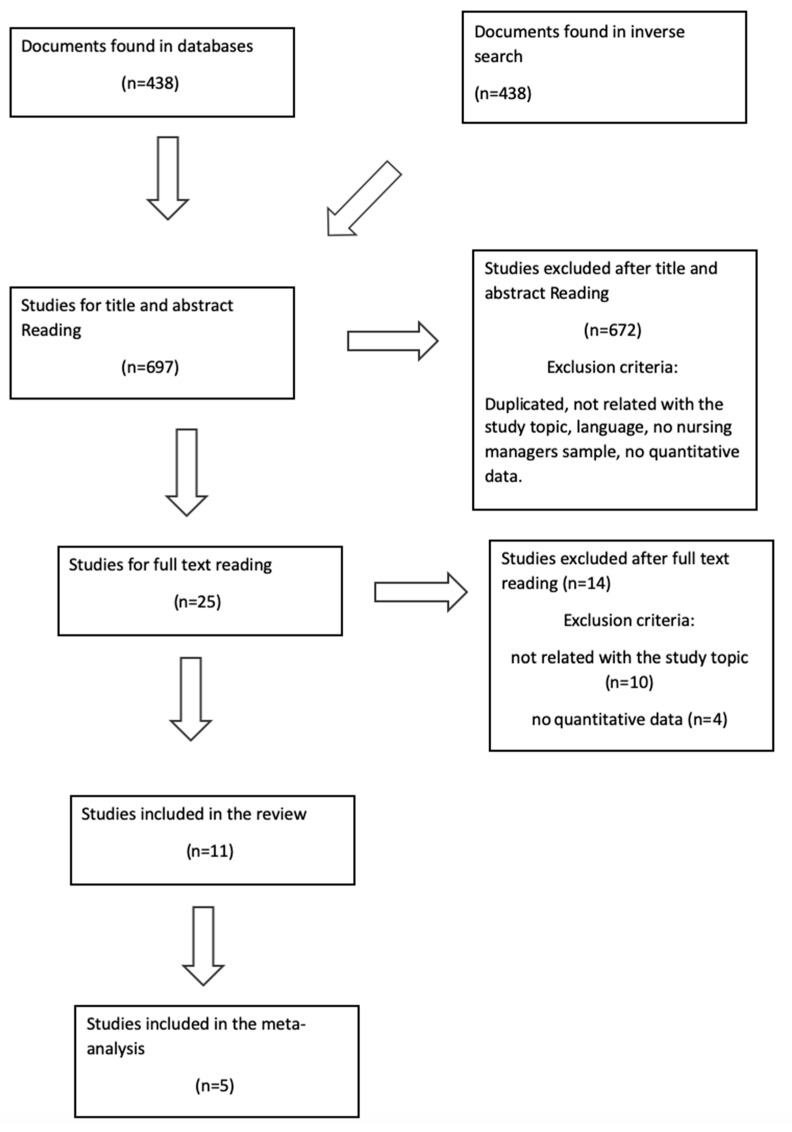
Flow diagram of selection process.

**Figure 2 ijerph-17-03983-f002:**
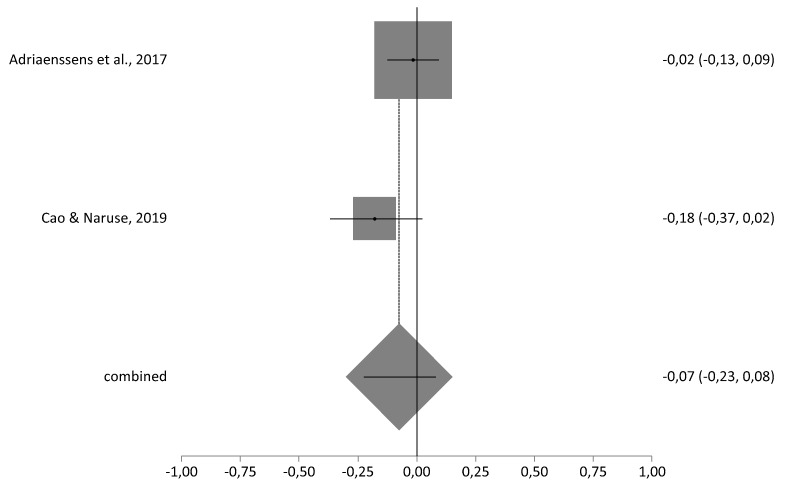
Forestplot of correlation between emotional exhaustion and age.

**Figure 3 ijerph-17-03983-f003:**
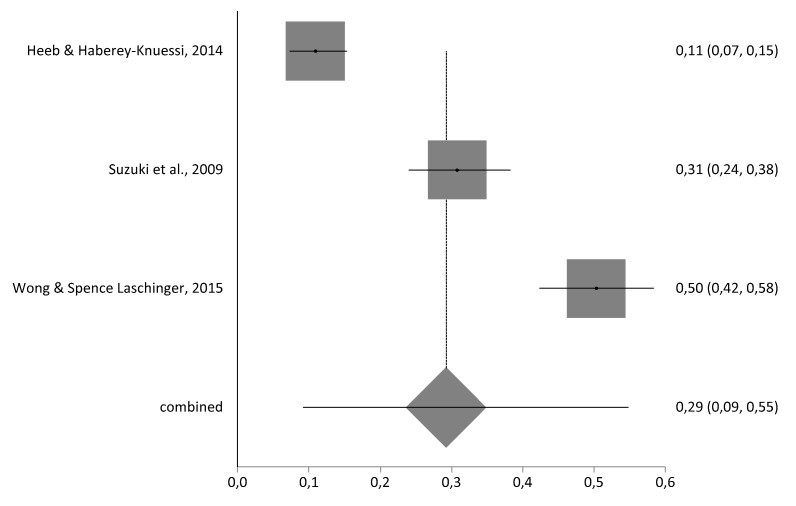
Forestplot of emotional exhaustion prevalence rate.

**Table 1 ijerph-17-03983-t001:** Main results included in the review.

Authors (Publication Year), Country.	Burnout Instrument	n	Characteristics	Mean Values (Standard Deviation)	Study Design/OCEBM*
Suzuki et al. (2009) [17], Japan.	MBIJ-RAS	172	Age: 43.8 ± 7.1 Married: 30.2% Children: 29.9%	Burnout total = 11.0 EE = 1.4 D = 3.6	Cross-sectional/ LE: 2C GR: B
Wong et al. (2015) [16], Canada.	MBI	159	Age: 48.1 ± 7.0 Female: 143 (92.3%) Male: 12 (7.7%)	EE = 2.91 (1.44) D = 1.58 (1.44)	Cross-sectional/ LE: 2C GR: B
Heeb et al. (2014) [18], Switzerland.	MBI-HSS	257	Age: 46.7 Female: 58.0% Married: 58% Children: 56.4%	EE = 16.9 (7.1) D = 5.4 (4.2) PA = 35.6 (6.9)	Cross-sectional/ LE: 2C GR: B
Spence Laschinger et al. (2008) [19], Canada.	MBI	134	Age: 48.04 ± 10.21 Female: 129 (95.3%) Male: 5% (4.7)	EE = 3.14 (1.15)	Cross-sectional/ LE: 2C GR: B
Adriaenssens et al. (2017) [9], Belgium.	MBI-HSS	319	Age: 45.7 (24–63) Female: 59.1% Children: 86.2% Full time: 85.8%	EE = 1.42 (0.71)	Cross-sectional/ LE: 2C GR: B
Kanste (2008) [20], Finland.	MBI-HSS	627	Female: 94% Male: 6% Age: 43.7 ± 9.0	EE = 2.95 D = 1.82 PA = 6.18	Cross-sectional/ LE: 2C GR: B
Cao et al. (2018) [15], Japan.	J-BI	93	Female: 97.8% Male: 2.2% Age: 43.26 ± 7.98 Full time: 67.7%	EE = 2.46 (0.91) D = 1.70 (0.70)	Cross-sectional/ LE: 2C GR: B
Lee et al. (1996) [21], USA.	MBI	78	Female: 93% Married: 79%	ND	Cross-sectional/ LE: 2C GR: B
Hewko et al. (2015) [22], Canada.	MBI	95	Female: 87 (92%) Male: 8 (8%) Age: 75% (40–59)	EE = 3.57 (1.38) D = 3.58 (1.50) PA = 5.08 (1.09)	Cross-sectional/ LE: 2C GR: B
Karsavuran et al. (2017) [23], Turkey.	MBI	244	Female: 78 (32%) Male: 166 (68%) Married: 75.4% Age: 38% (50s)	High scores for EE and D and low scores for PA were associated with high levels of burnout.	Cross-sectional/ LE: 2C GR: B
Ceravolo et al. (2019) [24], USA.	ProQOL CBI	12		The mindfulness intervention had a positive impact on the Professional Quality of Life and Copenhagen Burnout Inventory scores.	Longitudinal Prospective/ LE:2C GR: B

*Keynote:* EE [Emotional exhaustion]; D [Depersonalization]; PA [Personal accomplishment]; ND [No data]; OCEBM* [Levels of evidence of the Oxford Centre for Evidence-Based Medicine] [25]; LE [Level of evidence]; GR [Grade of recommendation].

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
