# Peer review of "Burnout in Nursing Managers: A Systematic Review and Meta-Analysis of Related Factors, Levels and Prevalence"

_ijerph, 2020, doi:10.3390/ijerph17113983_

Round 1
Reviewer 1 Report
This is a meta-analysis of 5 cross-sectional studies. The results of the study help to settle how burn out syndrome affects to nursing managers and their risk factors. I have several comments.
- Whether there have been similar studies in the past, what are the similarities and differences of burnout among nurses or doctors?
- Does the the databases used include Web of Science?
- Number of studies included in the mata-analysis was small (only 5).
- The number of different reasons of excluded studies.
- As to the Characteristics such as age , mean+- SD are commonly used.
- When analysis the correlation between burnout and age, is confounding factors excluded?
- When analysis the prevalence of burnout, how to intergrate different studies because they use different criteria?
Author Response
Dear reviewer,
Thank you for the review and for your comments. We think that our manuscript has improved with your suggestions. Please find below the response to each comment highlighted in yellow. All the changes in the manuscript have been tracked. Kind regards
This is a meta-analysis of 5 cross-sectional studies. The results of the study help to settle how burn out syndrome affects to nursing managers and their risk factors. I have several comments.
Whether there have been similar studies in the past, what are the similarities and differences of burnout among nurses or doctors?
Currently, there is few evidence about burnout in physician managers that we can compare with our results in nursing managers. More information about it has been included in the discussion.
Does the the databases used include Web of Science?
No, Web of Science was not used. However the used databases (Medline, PsycINFO, CINAHL, LILACS, Scielo and Scopus) cover most of the health science literature.
Number of studies included in the mata-analysis was small (only 5).
The number of studies included in the meta-analysis was small due to the low literature production about the topic in nursing managers. It has been included as a limitation of the study in the discussion.
The number of different reasons of excluded studies.
The number of different reasons of excluded studies has been included at the beginning of the results.
As to the Characteristics such as age , mean+- SD are commonly used.
All the studies have been checked and the information about age has been included in the manuscript following the format presented in the studies.
When analysis the correlation between burnout and age, is confounding factors excluded?
The confounding factors were not excluded due to the low number of studies included in that meta-analysis.
When analysis the prevalence of burnout, how to intergrate different studies because they use different criteria?
All the studies included in the meta-analysis used the same instrument for burnout measurement, the MBI. The MBI has cut-off points for low, medium and high levels of each dimension. We have included more information in the methodology.
Reviewer 2 Report
Well-written in general. Suggestions for English: US English uses "z" instead of "s" for words like analyze, minimize, organize. Might consider changing though I'm not sure what British English uses for these words: not a major problem either way.
Abstract: line 2: insert "to" after "was" and before "analyze"
Conclusion: 1st sentence: replace "settle" with "decribe" and remove "too" after "affects" and before "nursing".
2nd sentence replace: This meta-analysis paper is useful for recognizing mental health challenges of nursing managers and may increase awareness of the prevalence and impact of burnout in nursing managers. This should promote the prevention and treatment of burn out in these healthcare workers.
3rd sentence: Moreover, the manuscript, based on scientific evidence, promotes the use of new strategies to prevent burnout in nursing managers in the hospital environment.
line 246 insert "of" between detection and burn out
247 add "s" to characteristics similar to....
248 replace "as far as possible" with "as much as possible"
251 replace measures with "practices" to prevent the "development" of burnout
Note: be consistent with burnout or burn out
Author Response
Dear reviewer,
Thank you for the review and for your comments. We think that our manuscript has improved with your suggestions. Please find below the response to each comment highlighted in yellow. All the changes in the manuscript have been tracked. Kind regards
Well-written in general. Suggestions for English: US English uses "z" instead of "s" for words like analyze, minimize, organize. Might consider changing though I'm not sure what British English uses for these words: not a major problem either way.
Abstract: line 2: insert "to" after "was" and before "analyze"
The change has been done.
Conclusion: 1st sentence: replace "settle" with "decribe" and remove "too" after "affects" and before "nursing".
The change has been done.
2nd sentence replace: This meta-analysis paper is useful for recognizing mental health challenges of nursing managers and may increase awareness of the prevalence and impact of burnout in nursing managers. This should promote the prevention and treatment of burn out in these healthcare workers.
The change has been done.
3rd sentence: Moreover, the manuscript, based on scientific evidence, promotes the use of new strategies to prevent burnout in nursing managers in the hospital environment.
The change has been done.
line 246 insert "of" between detection and burn out
247 add "s" to characteristics similar to....
248 replace "as far as possible" with "as much as possible"
251 replace measures with "practices" to prevent the "development" of burnout
The changes have been done.
Note: be consistent with burnout or burn out
The text has been reviewed and now only says “burnout”
Reviewer 3 Report
The manuscript aims "to analyse the levels, risk factors and prevalence of burnout in nursing managers". Although the aim is relevant, there are some issues about definitions, methods and interpretation that are not deeply considered and brings doubts about how it was performed.
Title (throughout the paper): it is not clear what authors mean by "levels"; "burn out" or "burnout", both terms are found?
Abstract: why CI for correlation is presented as %?
Introduction: line 53 there is a reference inappropriately mentioned;
line 72 is not necessary
Methods: change the term "search equation" to string; at lines 82-83 exclude the phrase "thus understate publication bias"; at line 94: what was considered "low methodological quality"?
Results: at line 149 authors present results about risk and mediating factors. It would be nice to have their definitions at the Introduction; use . instead of , for decimal places; Figures 2 and 3 have a low definition.
Discussion: Authors explain a causal chain and preventive strategies that seem to be beyond the scope of their results.
Conclusion: are not solely based on the results.
Author Response
Dear reviewer,
Thank you for the review and for your comments. We think that our manuscript has improved with your suggestions. Please find below the response to each comment highlighted in yellow. All the changes in the manuscript have been tracked.
Kind regards
The manuscript aims "to analyse the levels, risk factors and prevalence of burnout in nursing managers". Although the aim is relevant, there are some issues about definitions, methods and interpretation that are not deeply considered and brings doubts about how it was performed.
Title (throughout the paper): it is not clear what authors mean by "levels"; "burn out" or "burnout", both terms are found?
The term has been unified to burnout. We talked about burnout levels because the MBI has three different levels in each dimensions (low, medium and high). It has been specified in the methods.
Abstract: why CI for correlation is presented as %?
It was a mistake, it has been changed.
Introduction: line 53 there is a reference inappropriately mentioned;
It has been modified.
line 72 is not necessary
It has been deleted.
Methods: change the term "search equation" to string;
It has been changed.
at lines 82-83 exclude the phrase "thus understate publication bias";
It has been changed.
at line 94: what was considered "low methodological quality"?
The information has been included in the methods.
Results: at line 149 authors present results about risk and mediating factors. It would be nice to have their definitions at the Introduction;
Some of the risk factors have been defined in the results because it was hard to define it in the introduction and give sense to those sentences in the introduction.
use . instead of , for decimal places;
. instead of , has been used for decimal places in the text. We cannot changes in the Figures because the figures are done by the program and cannot be modified.
Figures 2 and 3 have a low definition.
Figures 2 and 3 have been changed to TIFF format with higher definition.
Discussion: Authors explain a causal chain and preventive strategies that seem to be beyond the scope of their results.
The affirmations about those aspect in the discussion have been modified. There are now less categorical affirmations.
Conclusion: are not solely based on the results.
Some sentences have been removed to leave only the conclusions based on the results
Round 2
Reviewer 1 Report
The authors have addressed most of my concerns.
Author Response
Dear reviewer,
Thank you for reviewing the manuscript again and for your previous comments in Review 1.
Kind regards.
Reviewer 3 Report
Authors made changes in the manuscript. But, Figures 2 and 3 still have low quality and the term burnout appears as "burn out" and "burnout" throughout the text.
Author Response
Dear reviewer,
Thank you for reviewing the manuscript again.
The term burnout appears as "burn out" and "burnout" throughout the text.
We have checked the manuscript and we have found one time the word "burn out" in the conclusion. We have modified it to "burnout"
Figures 2 and 3 still have low quality
We have modified the quality of Figure 2 and 3 and now is higher.